

**A 'Mental Models' approach to the communication of subsurface**
**hydrology and hazards**
H. Gibson[1], I. Stewart[1], S. Pahl[2] and A. Stokes[1]
[1] *School of Geography, Earth and Environmental Sciences, Plymouth University,*
*Plymouth, UK*
[2] *School of Psychology, Plymouth University, Plymouth, UK*
Abstract
Communicating information about geological and hydrological hazards relies on
appropriately worded communications targeted at the needs of the audience. But
what are these needs and how does the geoscientist discern them? This paper
adopts a psychological 'mental models' approach to assess the public perception of
the geological subsurface and surveys three communities in the south-west of
England about their attitudes and representations of the geological subsurface. The
findings reveal important preconceptions and misconceptions regarding the impact of
hydrological systems and hazards on the geological subsurface, notably in terms of
the persistent conceptualisation of underground rivers and the inferred relations
between flooding and human activity. The study demonstrates how such mental
models can provide geoscientists with empirical, detailed and generalised data of
perceptions surrounding an issue, as well reveal unexpected outliers in perception
that they may not have considered relevant, but which nevertheless may locally
influence communication. Using this approach, researchers and communicators can
develop information messages that more directly engage local concerns and create
open engagement pathways based on dialogue, which in turn allow both groups to
come together and understand each other more effectively.
1 Introduction

Communicating geological information about geological and hydrological hazards
relies on appropriately worded communications (Liverman, 2010) targeted at the





needs of the audience (Nisbet, 2009). Those needs are often deemed to be what
geoscience professionals feel the public 'need to know', leading many hazard
messages to fall into the largely now rejected 'deficit model' of communication
(Sturgis and Allum, 2004). According to this model people need to educated about
those areas of knowledge in which they are seen to be deficient rather than taking
into account their existing knowledge structures and wider concerns or values.
Moreover, the responsibility for tailoring the communication to the target audience is
often placed on the public, requiring them to 'ask the right questions' (Rosenbaum
and Culshaw, 2003). This emphasis on the public's need to ask the right questions
misses a bigger issue in communicating geological hazards, which is the influence of
heuristics and bias in how people may interpret information, especially unfamiliar
scientific and technical data (Kunreuther and Slovic, 1996).

The subjective nature of risk communication and understanding in both experts and
non-experts is now well established (Slovic et al., 2004), but it is easy for risk
communicators to focus on improving access to information from the scientists'
perspective, and overlook the impact of experience- and emotion-based
preconceptions from the non-expert perspective (Leiserowitz, 2006). Commonplace
preconceptions will strongly influence the way that a non-specialist will access and
interpret the geoscience risk information provided to them (Liverman, 2010), and so
it is vital that public perceptions of geological and hydrological hazards are taken into
consideration by communicators.

The value in examining perceptions specifically is increasingly being recognised by
many in the risk communication community, including in disaster risk reduction and
commercial geology fields. Barclay et al (2008), for example, called for a more
interdisciplinary 'disaster reduction' approach to volcanic risk communication, which
includes stakeholders in policymaking, and uses social and physical science to work
together to produce more appropriate and effective communications based on the
needs of the community. Meeting the particular needs of at-risk communities through
collaboration between the physical and social sciences is now emerging as a fairly
central component of modern risk science (Donovan et al., 2012;Frewer, 2004;Lave
and Lave, 1991;Mabon et al., 2014).



It has long been known that when the public receives information, they can interpret
it - and therefore organise their reactions - in a variety of ways depending on their
perception of both the science and the scientist (Fischhoff 1995). Various inherent
cultural and social assumptions control the way that this information is interpreted
(Donovan, 2010;Mabon et al., 2014;Slovic et al., 2007). Thus, without examining a
population through social or psychological scientific inquiry, it is impossible to predict
how they will respond to a particular science communication message (Wynne,

75    1991).


A key challenge of communicating such messages, therefore, is that in addition to
the wider social or cultural impact on perception of scientific information, individuals
apply their own pre-existing ideas and concepts to any scientific data that they are
presented with (Mileti et al., 2004). In this context, psychology-based methods are
vital, and one such method is the 'mental models' approach (Morgan et al., 2002).
This paper outlines the mental models methodology and uses it to explore broadly
held perceptions of the geological subsurface, and from that examine how those
perceptions relate to geological and hydrological hazards.  Empirical evidence is
presented showing that such a method can provide valuable contextual data for
geological and hydrological hazard communicators.

2 Communicating Risk via Mental Models

Traditional views of risk communication have conventionally been based on how
best to align the knowledge of the recipient with that of the expert (or communicator).
Early work by Slovic (1987) demonstrated how several key factors underlie the
perception of risk in non-experts, such as 'familiarity' and 'dread' . A graphical
representation (Fig 1) shows the relative perceptions of different threats, as
organised by their varying degrees of familiarity and dread. The diagram shows that
certain threats, which may statistically be considered more risky – such as riding a
bicycle – are perceived to be far less risky than a statistically safer activity – such as
flying in a commercial aeroplane (Slovic, 1987). Later work coined the term 'affect
heuristic' to describe the important role of feelings in non-experts' risk assessments
(Slovic, 2010;Slovic et al., 2004).




103 *Figure 1. The perception of risk within a two factor space, representing public*

104 *perceptions of how risky an activity was based on its familiarity and how fatal the*

105 *consequences may be (Slovic, 1987 p98).*


107 The affect heuristic describes the way that an individual's perception can colour their

108 response to a piece of information about a subject, by ascribing greater or lesser

109 importance to the risk than an expert would, based on a logical assessment. The

110 affect heuristic can be described as a form of emotion, defined as positive or

111 negative feelings that are used to evaluate an external stimulus (Slovic et al., 2007).

112 The influence of heuristics and biases, such as the affect heuristic, is so central to

113 designing effective risk communication that more integrated methods of assessing

114 the public's perception of geological and hydrological issues need to be utilised

115 (Mabon et al., 2014).

116

117 By taking into account the impact of a non-experts' perception of risk, the field of risk

118 communication shifts from a one-way form of communication towards more of a

119 dialogue. However, even within this more inclusive mode of communication, an

120 outdated emphasis on the information and value judgements of the expert is still

121 apparent (Sturgis and Allum, 2004). By this account the 'top-down' transfer of

122 information provided by the expert must be translated by the emotional state of the

123 non-expert (Slovic et al., 2004) and integrated into their own 'lay knowledge' (Callon,

124 1999). Lay knowledge is generally dismissed as inappropriate by experts, who

125 expect decisions to be made on the basis of relevant technical information (Johnson,

126 2008), but there is growing acknowledgment of the role and value of individual and

127 community knowledge, not just in collecting and compiling scientific data (Lane et al.,

128 2011) but also in improving communications by countering the expert-imposed

129 concept of risk (Lave and Lave, 1991). One psychological approach that has been

130 employed effectively in communicating across a range of risky and controversial

131 geological and hydrological issues is 'mental models' (Lave and Lave, 1991;Maceda,

132 2009;Skarlatidou et al., 2012;Wagner, 2007;Thomas et al., 2015).


134 The mental models approach to communicating risk (Morgan et al., 2002) is based

135 upon the broader mental models theory, developed by Johnston Laird (1980) as a



conceptual paradigm that encompassed new ideas about language and perception
in the burgeoning field of cognitive science. The theory of mental models as
interpretation of theoretical reasoning has fallen from favour in psychology (Evans,
2002;Over, 2009), but it is still used in the applied sense, particularly by researchers
examining decision making associated with risk, communication and education
(Goel, 2007;Larson et al., 2012;Panagiotaki et al., 2009;Skarlatidou et al., 2012).

The mental models approach to risk communication is a form of deductive
reasoning, connected to decision making. The approach assumes that, in order to
make a decision about an issue, an individual will construct an artificial (mental)
reality in order to test a series of simulated scenarios using data previously collected
and valued by that individual (Morgan et al., 2002). The decision about what action
to take will be based upon a logical interpretation of the results of these tests, and
decisions are most easily made when the tests are simple (Johnson-Laird, 2013).

This method can be demonstrated by considering the decision of 'travelling down
stairs'. Whilst it may seem an exceedingly simple issue, by considering all the
different factors that might cause you to trip on the stairs and therefore what you may
have to do to control those factors, a researcher can build a model of what a person
considers when they are thinking of walking up or down stairs (Morgan et al., 2002).
This simple example, represented in Fig. 2, demonstrates the particular
effectiveness of mental models. In the diagram, some factors such as the floor
covering, lighting or the height and width of the stairs may be anticipated by experts,
and statistically assessed as being valuable factors to communicate hazards about.
The node that mentions 'sleeping habits of the cat' however may not have been
considered, and yet might be a key issue for a non-expert in this circumstance.

The use of mental models, therefore, allows the researcher to gain a better
understanding of the importance of many issues from both the expert and non-expert
perspective, and also allows for the inclusion of not just analytical reasoning, but
experiential as well (Leiserowitz, 2006).




*Figure 2. Illustration of the construction of an influence diagram for the risk of tripping and falling on the stairs: a) shows just those two elements; b) adds factors that could cause a person to trip; c) adds factors that might prevent a fall after a person trips; and d) introduces decisions that a person could make that would influence the probabilities of tripping and falling (Morgan et al., 2002 p37).*

In the context of geological hazards and risks, it was found that in cases where the risks are unfamiliar to the individual, mental models theory allowed the participant to explore the decision-making process more fully (Goel, 2007). When applied to specific contexts, most notably to radioactive waste management and carbon capture and storage (Skarlatidou et al., 2012;Vari, 2004;Wallquist et al., 2010), it was found that in cases where the perceived risk of new technology was greater than the actual risk (or the risk designated by the expert), mental models provided a useful holistic approach to decision making, that placed equal value on the attitudes of both expert and non-expert (Vari, 2004).

An important aspect of the mental models approach is in the value placed on the data coming from the non-expert. In placing the non-expert in a position of equal authority with the expert, any information provided is also represented as equally important (Morgan et al., 2002). This draws the communicator away from the one-directional deficit model of communications (Bucchi, 2008) and towards a more dialogic model, where the perceptions of the non-expert are not simply misconceptions to be adjusted, but instead become concerns to be addressed through discussion and interaction. The approach allows researchers to assess not only what participants (both expert and non-expert) involved with an issue think, but also why they think it (Kiker et al., 2005). This is valuable to both expert and non-expert, as it allows both parties to fully express their perceptions of an issue and come to a greater understanding of the other party's perspective. The approach therefore allows the refinement of communication to focus on messages that are salient to both communicator and recipient, which will increase the efficacy and significance of these communications (Frewer, 2004).

3 Applying the Mental Models Method



The mental models approach to risk (Morgan et al., 2002) is a mixed method
procedure which integrates aspects of Johnson-Laird's Mental Models theory (1983)
with risk communication practice (Morgan et al., 2002). It assumes that the heuristics
and biases used by non-experts to interpret controversial, critical or unfamiliar issues
do not form an entire model that directly reflects the world as the participant
experiences it, but rather constitute a series of interconnecting ideas that may colour
the perception of an issue (Morgan et al., 2002). This qualitative and quantitative
process consists of three main stages:

1. Qualitative semi-structured interviews are conducted with a broad sample of

the target population, as well as with technical experts in the field under

question. These semi-structured interviews provide the participant with an

opportunity to speak freely about the issue, but also discuss related or

perhaps peripheral topics that the participant feels is relevant (Mabon et al.,

2014). Once this stage is completed, a series of models are constructed

which reflect the key perceptions held by each group and considers how

these perceptions compare across groups of different 'expertise'.

2. Quantitative questionnaires are constructed from the models produced after

the interview stage. These questionnaires test the dominant perceptions that

are highlighted by the model as representing the area of greatest concern or

interest for the participants and researcher. The statements or questions are

constructed using the language of the participants so as to minimise bias. The

results of the questionnaire are then compared to the original models to test

their validity in a larger sample.

3. If the model provides a good reflection of the dominant perceptions of the

target population, then a communication is designed that dovetails with the

model content, in order to stimulate useful dialogue or provide information.

This communication is tested for its ability to improve knowledge and

understanding in the target population.


In this study, Morgan et al's (2002) approach was combined with a three dimensional
(3D) participatory model during the semi-structured interview stage. The use of a 3D
participatory component, whereby participants either use or create a 3D model in the
elicitation process, reflects the recognition that often participants in an interview may





have difficulty expressing their thoughts verbally in an interview (Cooke and
McDonald, 1986;Ongena and Dijkstra, 2007). Because geology is a very descriptive
and visual science (Frodeman, 1995), this can lead to misinterpretation of ideas from
both the expert and the non-expert. To address this issue, previous studies of
geological risk have employed 3D participatory modelling to provide an alternate
method of elicitation during focus groups or interviews (Cadag and Gaillard, 2012).
The inclusion of the 3D model provided participants with a means to test their
verbally expressed concepts in an alternative format.

4 Details of present research and research questions

This study presented in this paper represents a part of broader research into what
perceptions people hold about the geological subsurface. This broader study
covered all aspects of a society's interactions with geology including: industry,
heritage, and recreation. The present analysis focused on a subset of issues
particularly relevant to hydrological interactions with the subsurface environment and
the hazards that this might influence. This research examined common ideas and
attitudes to the subsurface with reference to how experts and non-experts
conceptualise the geological subsurface. In particular, questions were addressed
that included: conceptualisation of the structure of the subsurface environment, the
impact of human activity, and the influence of natural forces or phenomena.

A combination of participatory, qualitative and quantitative methods was used. The
3D model comprised a 1m x 1m x 1 m sized whiteboard cube, on the top surface of
which was a topographically-moulded aerial photo of each study location. The aim
was to enable participants to visually represent those concepts that related to the
subsurface environment in their area.
Interviews were conducted by the primary researcher (H.G.) - a geologist with
practical experience working as a formal and non-formal science communicator in a
museum and national park. Care was taken by the researcher to limit bias during the
interviews and a conversational protocol (a relaxed back-and-forth conversational
style) was employed during the interviews (Ongena and Dijkstra, 2007).



Three locations were selected for the purposes of the survey: one village in Cornwall
and two villages in Devon. These villages had similar demographics - as assessed
using the 2011 census data (Office of National Statistics, 2011) - but different
exposures to geology. The first village, Carharrack in Cornwall (population 1324),
has a strong cultural and historical association with geology, but little current
geoscience activity in the immediate proximity. The second village, Sparkwell
(including Hemerdon) in Devon (population 1246), has a moderate cultural and
historical association with geology, but has a prominent current geological industry
active in the immediate vicinity. The third village, Chulmleigh in Devon (population
1308), has neither a strong cultural and historical association, nor a current
geological presence; indeed the local geology is not particularly visible in the
landscape.

The study incorporated both expert and non-expert interviews. Six interviews with
experts (individuals with considerable experience either in the academic or industrial
side of geology local to the area under survey) were conducted as well as a literature
review of data relevant to a non-expert's understanding of the subsurface. Non-
expert participants were selected using a 'snowball' method (Forrester, 2010) after
initial contact with parish councils was made to establish local awareness of the
study and paper adverts were placed in prominent locations around each village.

A total of 29 interviews were conducted across the three sites. Interviews continued
until a broad sample was achieved and repetition of concepts between participants
occurred (Morgan et al., 2002). In line with the ethical approval granted by the
University of Plymouth Science and Technology Ethical Committee, the names of all
participants have been anonymised and replaced with factious names as is
demonstrated in the results section. The interviews were conducted between
January and September 2014. The questionnaire was distributed by post to all
households (5214) in the target areas during September 2015, with a total response
rate of 228 (4.37%) both online and through the mail. During the time of the initial
interviews the UK was experiencing unusually severe winter storms that resulted in
damage to key infrastructure across the southwest (e.g. disruption of main Devon-
Cornwall rail line at Dawlish), extensive flooding and some loss of life. This high-





profile flooding may have influenced the content of the interviews, especially those
conducted between January and March 2014.

5 Results: Perceptions of the subsurface, water and geological hazards from 3D
drawings

Participant responses to the semi-structured interviews were diverse and
represented a wide range of opinions and perceptions. Although detailed mental
modelling of the full set of responses is ongoing, an analysis of a subsection of the
results allows some provisional observations to be made.

The main attention of the study was focussed on the geological subsurface, so first
this paper will provide context with some generalised results about the subsurface
using the data collected with the 3D participatory models. These models provided an
insight into how people visualise the subsurface environment in their area, and in
combination with the verbal results, provide an interesting idea of the perceptions of
the subsurface the people in these three villages hold.

As experts and non-experts participated in interviews with the same structure and
substance, their results can be directly contrasted to highlight similarities and
differences. The images in Fig. 3 demonstrate some of the key concepts
demonstrated by participants.


*Figure 3: Images of 3D participatory models completed by expert and non-expert*
*participants. a) Eric – an expert participant, represents the expert model, with a*
*logical diagram utilizing more than one side of the model (including the surface), with*
*detail provided by numerical and factual annotation. b) Edward – an expert*
*participant, also demonstrates an expert model, with a representation of a fault*
*structure displayed at the surface and symbols used to identify different rock types.*
*c) Kimberley – a non-expert participant from Carharrack, conceives the subsurface in*
*a couple of interesting ways. Firstly, the red shading is used to depict the Earth's*
*core, initially as a semi-circular shape and then later modified to match the linear*
*appearance of the rest of the diagram. In addition, the diagram shows some*



*uncertainty about the inferred ground level, which is drawn with a green zigzag line,*
*below the actual surface of the model. d) Katie – a non-expert participant from*
*Carharrack, presents a much sparser diagram, with subterranean buildings*
*emphasizing the human interaction with subsurface space. e) Charlotte – a non-*
*expert participant from Chulmleigh, drew a direct link between the surface and the*
*subsurface in the form of a channel that connects the topographic low (where the*
*river is shown on the aerial photograph) and an underground body of water, which*
*cuts across the entire model. Finally, f) Charles – a non-expert participant from*
*Chulmleigh shows another model which has been interpreted to represent a more*
*scientific model, with the Earth's core represented at the bottom and the different*
*layers as being approximations of different scales of geological concepts, from*
*tectonic plates to erosional surfaces of sandstone.*

**5.1 General perceptions of the subsurface from 3D model verbal explanations**
One of the initial observations was in the application of 3D spatial reasoning by the
experts. This is clearly visible in Fig. 3a and Fig. 3b, where both Eric and Edward
utilised more than one side of the model in association, as well as making reference
to the surface image for contextual cues. The use of 3D spatial reasoning was
common throughout the expert interviews, as this comment from Ethan indicates:

…so as you go down this could be all killas[1], and could be cut off

by…by… you've got lots of joints, so you have footwalls and hanging

wall and slip planes. So you could find that down here, the further you

go away from the hill, you find the granite's further away?

Ethan, commercial and academic expert


This description includes an inherent use of 3D spatial reasoning, demonstrated by
Ethan in his inference of a change in location of the granite relative to the hill as
influenced by the joints and slip planes. In general it was clear from the way that the
experts used the block models that they were using 3D spatial reasoning. There was a
deliberate connection made between the adjacent walls of the model cube, and also
with the surface topography and the aerial photograph. The experts completed the

---

[1] A regional term for Devonian-Carboniferous low grade phyllite (Kearey, 1996)





models with a great deal of gestural explanation (Kastens et al., 2008), even to the
extent of using the pens provided for annotation to demonstrate a fault structure
present in the area (visible in Fig. 3b).  This 3D spatial reasoning was not, however,
present to the same degree in the non-expert participants. Some spatial reasoning
was used, but it was most often utilised in a purely geographic two dimensional way.
Moreover, all of the non-experts limited their elicitation to a single side of the model
cube.

I'm surprised really that that [the quarry] is in a quite high part

compared with others.  As you move down here, I know from my own

experience, as you come south from here, the fall of the land is down

here and the rocks are actually a bit softer from my experience.

Henry, Hemerdon and Sparkwell


The models also demonstrated another consistent difference between the experts
and the non-experts, and that was an anthropocentric, or human focussed view of
the subsurface (Slovic, 2010). Whereas, for the expert participants, geological
activity was considered a product of the local geology, for many non-expert
participants, human interaction with the subsurface was the only important factor.
This anthropocentric perspective of the subsurface is demonstrated in Fig. 3d, which
also indicates how some participants who held a strongly anthropocentric model had
a great deal of difficult in adding any other detail to their expressed perception of the
subsurface.

Q: So, if you were to, like, dig straight down now, what would you

come across?

395          A:  I don't know.  I don't want to know.......There could be things

underneath the ground like that kind of thing.... Other houses, I don't

know.

Katie, Carharrack


Perceptions shaped around human concerns contrast with the more expected
conventional geological depiction of subsurface relations (e.g. Fig. 3c). These types
of diagram (called 'scientific' from here on) varied in the level of detail provided, with



some (Fig. 3e) being very detailed, and exhibiting a large amount of additional
annotation relating to dates and eras, both historical and geological. These non-
expert scientific models focus attention on a range of themes. Some participants, for
example as shown in Fig. 3c and Fig. 3f, focus very strongly on the centre of the
Earth. In Fig. 3f the focus was more specifically related to the types of layers one
might encounter if penetrating the subsurface, but also included a visual link to the
Earth's core, which was identified early in the construction of the diagram. The role
and importance of underground water was also indicated in the way that participants
depicted the subsurface, such as with rounded pebbles.

A key point emerging from the semi-structured interviews was a strong
disassociation between the subsurface and the surface environment in non-experts.
This is most evident in Fig. 3c, where despite the surface of the cube being a
representation of the topographic surface, and the respondent being asked to
present what she thought was 'directly beneath her', an artificial ground surface was
added. This disconnection was demonstrated in multiple model depictions and,
alongside the limited use of 3D spatial reasoning, is a strong discriminator between
the non-experts and the experts.

When a connection between the surface and subsurface was presented by non-
experts it was frequently vague and portrayed in a general sense that was more
related to the nature of the rock in the area, as is evident in the following quote:

But granite, I would have thought, just about everywhere, really.  I

don't know what depth that would be.  It's probably near the surface

but I would have thought there would be granite around.

Katrina, Carharrack


In this example, the existence of a particular rock type was not consciously linked with
any visible landscape feature.  In contrast, the remarks below highlight an expert
connecting a mapped unit of geology below with a specific landscape feature above,
and using the observable outcrops as cues to discern the underlying differences in
local geology.


Well perhaps it's not the same sandstone for a start, you can make a
measurement of one sandstone in one hill there and then you know
it's dipping towards the hill, …er …towards us, and because that
sandstone is all the same, it could be a completely different
sandstone.

Edgar, geoscience expert


**5.2 Combined mental model**

By integrating the findings of experts and non-experts from the three study areas, a
final combined mental model has been obtained (Fig. 4). This model represents a
collective view of the public perception of the geological subsurface, especially
focusing on the interaction between surface and subsurface elements in this
conception. The central feature is the connection between the surface and the
subsurface. Most participants alluded to some degree of linkage, but it was the expert
participants who consistently used this connection in constructing their subsurface
model. This difference between the experts and the non-experts was also present in
other shared nodes, such as 'layers' and the 'soil-rock boundary', but of particular
interest to this study is the emphasis from the non-experts on the nodes of 'water' and
'flooding'.


*Figure 4. A mental model of expert and non-expert perceptions of the subsurface in*
*the southwest of England. Rectangular nodes are those shared between experts and*
*non-experts, oval nodes are those expressed by non-experts alone. The three*
*frames '3D thinking', 'scale' and 'technical and local terms' have been placed*
*externally as they provide context for all of the other nodes.*

6 Detailed analysis of themes relevant to Hydrology and Hazard

To explore the usefulness of this model for applied geoscience in general and
geohazards in particular, this section examines in more detail the two non-expert
nodes, 'water' and 'flooding'. These nodes potentially offer an interesting insight into
the general perceptions of the non-experts into the geological subsurface. Both



relevant data from the 3D participatory approach and the larger survey will be
reported here.


**6.1 Underground rivers.**
Firstly, although water was mentioned by the expert participants, it was very much a
peripheral concept, more closely related to mining activities and industry.

We'll have to satisfy the Mines Inspectorate that what we are doing is

safe and won't result in potential mine flooding. So …er…I don't know,

I suspect that the …er… presence of those mine workings would be a

nuisance if we drilled into them so we have to avoid them from that

point of view, but potentially represent quite a good…er…water

source for us.

Eric, commercial geology expert


For the non-experts, however, the presence and movement of water was frequently
mentioned, most prominently in the recurring notion of underground rivers.

I think you'd find a lot of water and I imagine there would be lots of

channels. Cos I think the water would have to seep into the ground

and it has to run down cos we are so high that I think there would be

an underground network of holes or natural sewers.

Just because of the pure volume of water that we have and we don't

flood as much so there might be some kind of water table that bits of

land, kind of, not floating on top but almost like resting on top.

Christian, Chulmleigh

I think water, if you go down, there's... you know… water would

come off of different bits, different directions and little bits, a bit like

underground streams really, but then finally I think you'd get these

solid stones where there's nothing there really.

Charlotte, Chulmleigh






Well, I think water, you know, the amount of rain that we've had you
know, over the last couple of years especially, it's not better for this
area… [Laughter] …because it gets into these tunnels sometimes I
think and then it…just got nowhere to go.

508                                        Kim, Carharrack


So I imagine that the top, the top sort of surface, would be 15 feet,
and then you would get into a granite and that would be, I don't know
how far down then. That would go on down and I imagine that in that
there are waterways and underground streams and that sort of thing.
Going through the granite.

515                                  Howard, Hemerdon and Sparkwell


The existence of underground rivers as the principal pathway for water to move in
the geological subsurface was so common that one of the questions in the
subsequent questionnaire was dedicated to it. Questionnaire recipients were asked
how much they agree with the statement: 'Water naturally forms channels
underground in order to flow through rock'. The majority of respondents (78.9%)
chose to either agree or strongly agree (Fig. 5.), showing how prevalent this
perception was amongst the questionnaire sample population.


*Figure 5. Attitudes of questionnaire respondents (n=223) to the statement 'Water*
*naturally forms channels underground in order to flow through rock'.*

This misconception of subsurface water routeways also appeared to relate to the
permeability of water through different rock types. Some types of rock seemed to be
perceived as allowing water to pass through them more easily, but other types of
material such as clay were more of a barrier.

But, a lot of it must be broken killas underneath because it - water -
literally drains, disappears. You don't get waterlogged ground
generally in this area, you know.

537                                        Kenneth, Carharrack





So there is water under us here which I suppose has been formed or
collected in certain layers - or runs through certain geological layers,
but right under this house - or under Chulmleigh, I couldn't tell
whether we were built on rock or what sort of strata, to be honest.
There's a lot of stone, I wouldn't have thought it's granite but it could
be.
545                 Christopher, Chulmleigh

**6.2 Water moving through rocks.**
Some participants also attempted to explain how water does move through rocks,
with particularly descriptive techniques.
I think it filters through the rock. Yeah, I think it does. It comes down
like rain through rock, doesn't it? And as long as they're pumping,
then they've got a dry place to work, but it will come up as it did until
the mine floods.  And I think it will flood almost to surface, as far as I
remember.
556                 Kara, Carharrack

When this notion of the permeability of rocks was posed in the questionnaire as
'Water cannot flow through solid rock' (Fig. 6), the majority of respondents answered
the question correctly, agreeing that water could pass through solid rock (although
many added an additional note to the question specifying different types). Just over a
fifth of respondents, however, selected the 'don't know' option (as well as eight
participants who left the answer blank), which suggests a significant level of
uncertainty exists in public perception of subsurface hydrology.
*Figure 6 Attitudes of questionnaire respondents (n=220) to the statement 'Water*
*cannot flow through solid rock'.*
**6.3 Water and instability.**





Another common concern expressed by participants was that presence of water in
the subsurface would result in instability and possibly cause ground failure or
collapse. This notion was expressed differently in the different locations. In
Carharrack, for example, the sense of instability was strongly connected to the
historical mining heritage present in the area.
It's a different kettle of fish mind you those sinkholes, but I'm
wondering if a lot of rain is seeping into old mine workings and might
make them sink.
580                                 Kevin, Carharrack
In Hemerdon and Sparkwell, in contrast, concern was expressed for the impact of
new mining activity on existing hydrological environments.
You can't keep digging up what's underneath you. It alters things. It
alters the landscape. It alters what comes out of the ground. It alters
the water table.
588                     Hannah, Hemerdon and Sparkwell
Finally, in Chulmleigh instability was expressed in relation to erosion – particularly of
arable land - which was often also connected to flooding.
We were on the point where the river comes right through and we
noticed that the river was taking away part of our land so I called in
somebody to explain that rivers do that, they change course and
lose some and you gain some…. But we didn't get flooded; it wasn't
a question of that, just watching my land being washed away and
deposited on somebody else's land.
599                       Chester, Chulmleigh
For the experts, this connection between geology and flooding had been a fairly
logical one, but in general non-expert participants did not consider this issue a
geological link. Instead, most believed that the flooding had a definite cause and it
was connected to human activity on the floodplains.






Q: Can you think of anything you've seen to do with geology in the
news recently?
A: No, except…um... and this is a bit broad, the flooding in the
Somerset Levels and that's not…really... to do with that [geology].

610                                        Christie, Chulmleigh


So much of things I think of relate to geography I suppose, whether
it's flooding in Bangladesh or India or China you know so it's more
geography related rather than geology. I'm not sure it contributes.

615                      Heather, Hemerdon and Sparkwell


I know you have to progress [with new mining development]. To
what end, though? Because you can keep progressing and now look
at us. We're getting all this flooding.

620                      Hannah, Hemerdon and Sparkwell


Although attitudes to flooding and ground instability caused by the presence of water
were not investigated directly, the evidence from the qualitative interviews provides
interesting inferences. The non-expert misconception of underground rivers was not
anticipated at the outset of the research, although it could possibly be expected from
anecdotal experience (Meyer, 1987). Common misconceptions like the prevalence of
underground rivers expose deeper issues, such as the public's understanding of how
water moves through subsurface environment and how water in the subsurface can
impact ground stability (Thomas et al., 2015).

**6.4 Additional/other themes.**
This type of study also provides useful context for communicators. For one thing, the
qualitative interviews themselves show the value that the public place on gaining
new and more detailed information that will allow them to continue to make effective
decisions about our changing environment.

And actually, I have to say the Somerset levels recently have
made me think a lot more about the geology and how they flood



and how we build on floodplains. We're taking no notice of what's
underneath and whether anything can drain away. So, I think it
would be much more important to all of us soon.

642                           Kimberley, Carharrack


As well as 'making public' misconceived ideas about how the natural world works,
mental models can expose non-expert perceptions that are so outlandish that the
expert might never have considered them. In the following statement, a non-expert
links news stories he has heard about earthquakes and fracking with resource
extraction.

It does concern me a bit sometimes the number of major
earthquakes we seem to be getting around the Pacific. I'm
wondering why. Is it something we're doing to the world that's
causing this? I don't think its fracking because they aren't fracking
there. Maybe because they're taking oil out of the ground and its
releasing pressure so that the world plates can move about a bit
more. I don't know.

657                   Hugh, Hemerdon and Sparkwell



7 Conclusion

Beyond the occasional ability to expose fairly perverse misconceptions about the
Earth systems, the mental models approach provides valuable context for
geoscience communicators. Its main benefit lies in bringing to light alternative
scenarios that are central to the way some participants' analyse the processes that
operate beneath their feet. In this regard, the heightened 'anthropocentric view' is an
important perspective, and one that has been recognised previously. Lave and Lave
(1991), for example, found in a similar study that some participants would orientate
their whole perception of past and future flood events on the fact that they were
'human-made'. Not appreciating the geological aspects of flooding may mean that
people conceive an inaccurate view of local flooding threat (e.g. from rising
groundwater levels).




Ordinary people's anthropocentric depiction of the subsurface is likely to have been
overlooked by communicators because it is not present in the expert interviews in
any noticeable way. It is revealed because the mental models method establishes
direct comparisons of expert and non-expert perceptions on the same issue. Such
inter-comparisons highlight fundamental mismatches of thinking, such as the use of
3D spatial reasoning and the logical connection between the surface and the
subsurface. They also shed light on the reasoning behind misconceptions, such as
the ubiquitous popular references to underground rivers, and offer up additional
nuanced detail to communicators attempting to grasp the public viewpoint.

Through mental models, geoscientists can be armed with empirical, detailed and
generalised data of perceptions surrounding an issue, as well as being aware of
unexpected outliers in perception that they may not have considered relevant but
which nevertheless may locally influence communication. Using this approach,
researchers and communicators can develop information messages that more
directly engage local concerns and create open engagement pathways based on
dialogue, which in turn allow both groups to come together and understand each
other more effectively. Given the ongoing wider challenges in geoscience
communication, especially in contested subsurface interventions associated with
shale gas extraction, carbon capture and storage and radioactive waste disposal, the
ability for geo-communicators to be carefully attuned to how individuals and
communities think will become ever more severely tested.

Author Contributions
H. Gibson, I.Stewart and S.Pahl designed the survey protocols and interview
questions. H.Gibson conducted all interviews and completed primary analysis and
construction of the mental model. A.Stokes and S.Pahl assisted with secondary
analysis of data and construction of mental model. I.Stewart assisted with
construction of the mental model. H.Gibson designed the questionnaire with
assistance from S.Pahl, I.Stewart and A.Stokes. H.Gibson prepared the manuscript
with assistance from all co-authors.

Acknowledgements





This work was supported by the Natural Environment Research Council (Quota
Award number 236443). The authors would like to thank Dr Mark Anderson for his
assistance and supervision during this research. The authors would also like to
acknowledge the valuable support of Robert Collier, Marine School Plymouth
University, for his assistance in the construction of the 3D participatory models.

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


Figures:

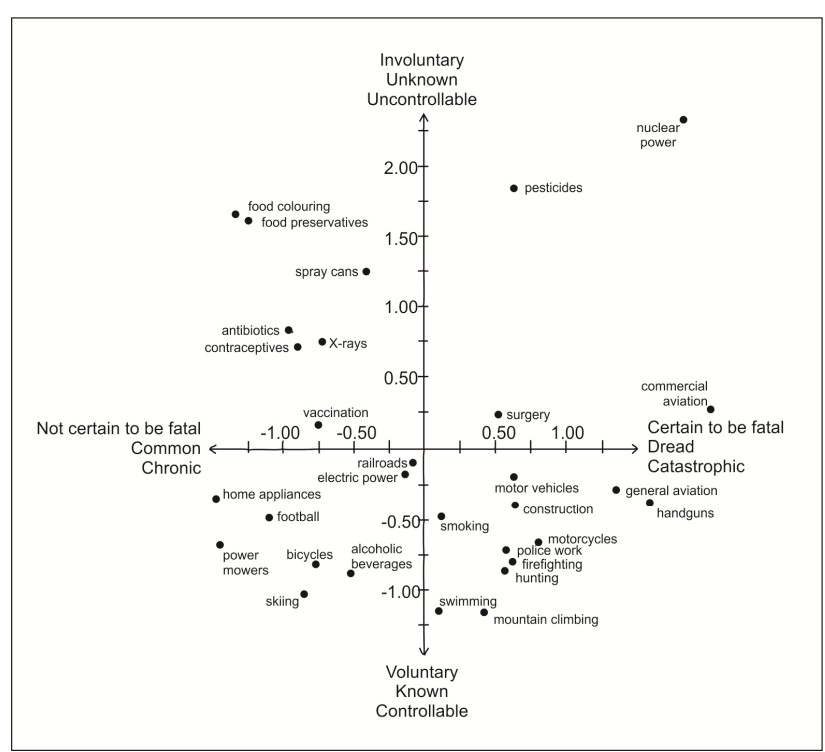


Figure 1.





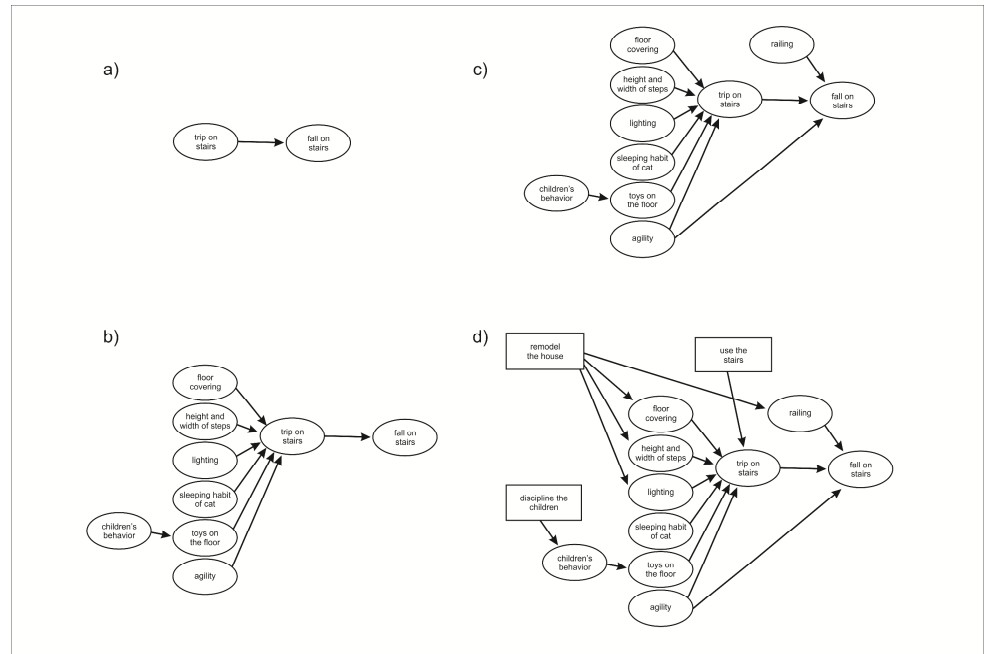


Figure 2.






Figure 3.




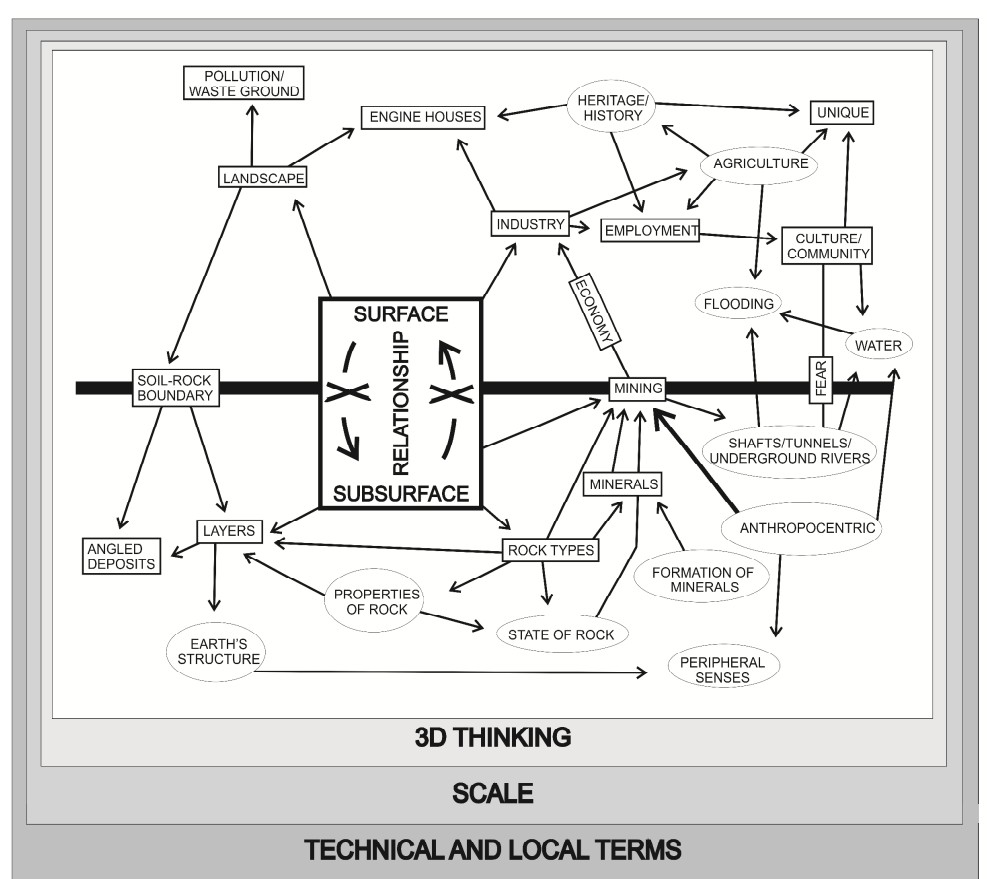

Figure 4.

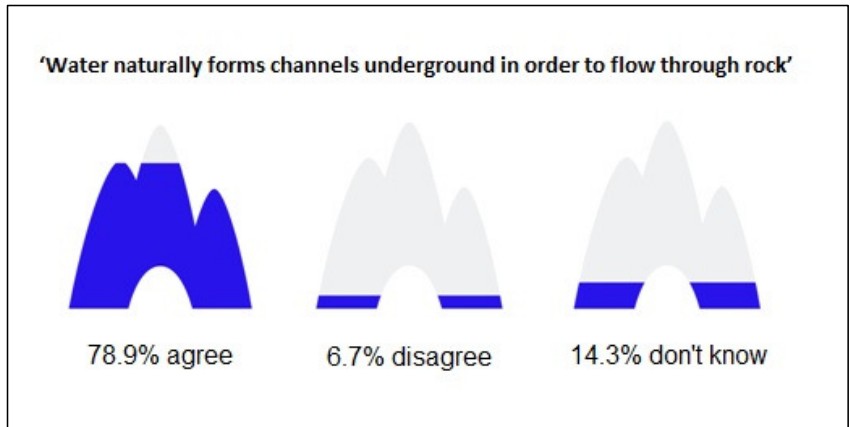

Figure 5.




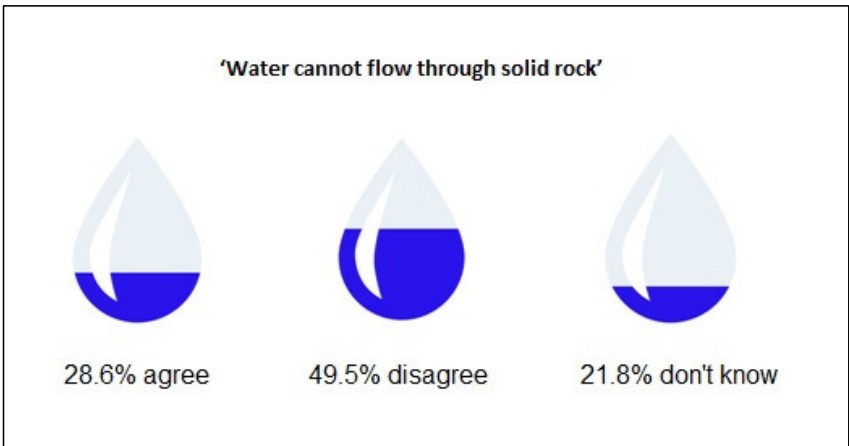


Figure 6.