# Peer review of "A 'Mental Models' approach to the communication of subsurface hydrology and hazards"

_Hydrology and Earth System Sciences, 2015_

## Referee Comment (RC1) · Anonymous Referee #1 · 26 Feb 2016

Evaluation:

The study by Gibson et al. describes a really interesting approach to geoscience communication, one which I have little seen in geoscience journal articles to date. The outcomes from the study are both insightful and useful to practicing geoscience 'experts' to understand the potential pre-conceptions of a 'non-expert'. As someone with experience in attempting to communicate contested geoscience topics to the non-expert, the results of this study highlight how failing to consider these pre-conceptions will continue to cause miscommunication when exploring controversial issues. My view is that there are currently few studies out there that geoscientists can readily understand from the work of social science, and this is a great start to moving towards better designed communications. This study will definitely impact on the way I personally communicate and hopefully others too, by introducing specific concepts first before dealing with more

complex issues.

As a geoscientist reading this paper, with limited understand of social science methods, certain concepts and phrases could benefit from further clarification (see comments). The manuscript may also benefit from some practical conclusions about how these methods can be used by geoscientists in the situations we often find ourselves in; for example public meetings trying to explain a topic that the 'non-experts' have little background knowledge of.

Based on my reading of this work I recommend it be accepted pending minor revisions.

General comments:

1. The manuscript is well organised and for the most part clearly written, with some language inaccessible to those not from a social science background.

2. In the introductory section, would it be possible to give an example of how poor communication can lead to a misinterpretation i.e. people's perceptions of underground rivers and how this leads to a certain perception of risk related to flooding.

3. The example of a mental model ('travelling down the stairs') is confusing – who are the 'expert' and 'non-expert' in this case? Is the expert someone who lives there (who knows they have a cat) and is the non-expert someone who is visiting and doesn't know they have a cat? Is it the case that the non-expert could fall over the cat because the expert may not have thought to mention it? Needs clarification.

4. Conclusions – how can this work for other geoscientists who may not have time to carry out the whole process?

Technical corrections:

P1 Line 32: Delete first 'geological'?

P2 Line 44: A sentence on the definition of 'heuristics' would be beneficial for geoscientists.

P2 Line 77: Could you give an example of what 'such messages' might be?

P7 Line 212: Are these semi-structured interviews done one on one?

P7 Line 224: Are there multiple questionnaires produced from the interviews, one for the expert and one for the non-expert?

P8 Line 252: Why did you choose hydrological interactions? Was this highlighted as something to look deeper into after the initial interviews?

P8 Line 260: Amend spacing's between last '1m'.

P9 Line 287: Could you very briefly explain the 'snowball method'?

P13 Line 403: Fig. 3e, do you mean Fig 3c? I can't see any additional annotation on 3e.

P16 Line 529: 'Permeability of water through different rock types' – confusing sentence, makes it sound like the water is permeable, not the formation. Please re-arrange.

Figure 1: Could you highlight the activities mentioned in the text so they're easy to find on the graph, they're difficult to find. And potentially shade in the separate zones to make them clearer.

Figure 3: It might be helpful to label the images expert or non-expert.

---

## Referee Comment (RC2) · Anonymous Referee #2 · 6 Mar 2016

Summary: The goal of this manuscript is to explore the power of "mental models" in revealing perceptions surrounding an issue, in this case study geological and hydrological hazards. The study has good motivations given the increasing cost of natural hazards or environmental change. The authors present a mental model of expert and non-expert perceptions of the subsurface the three communities in southwest England. Semi-structured interviews conducted with experts and non-experts in these communities revealed for the authors that non-experts exhibit a strong anthropocentric view in their perception of the environment and associated perceptions. The results and discussion section of the study is clear and very informative, however, I find the introduction and background section repetitive and not well structured. In the first part of the introduction the authors argue that communication of hazards is influenced by the heuristics and bias of how people perceive and interpret information and that without

any social or psychological scientific inquiry it is impossible to predict how people will respond. I am somewhat surprised by this statement because people of the age of 60 or younger should have received basic education in earth sciences in middle or high school, thus, there is often some base level of knowledge that scientists can relate to.

The authors state that people often apply their own pre-existing ideas or concepts to scientific data. I would argue that many people use analogies to translate scientific information to their own life-situations. Yet the concept of analogy does not seem to be a part of the mental model nor is it mentioned in the study (e.g. lines 68-75).

I disagree that lay knowledge is generally dismissed as inappropriate by experts and would like to see an explanation for this statement. In applied sciences and outreach my experience is that lay knowledge is quite often used to produce connections and analogies between established or believed science knowledge that is often deep-rooted in the public and new/shifted conceptions of science that experts try to convey. Often it is the only avenue that scientists have and can relate to in people to cause a change of mind. I would argue that deductive reasoning does not always play a role in the decision making, in particular if decisions have to be made about high risk topics such as natural hazards (e.g. Evans 2003, Trends in cognitive science; Darlow & Sloman 2010 WIREs Cogn Sc). Intuition and deliberation that are directly tied to the perception of objects and events (e.g., fear) can be powerful mechanisms as well (e.g. lines 143-149).

Please provide more information on how qualitative semi-structured interviews and the quantitative questionnaires are designed. How and by whom are questions for these interviews designed. How do you ensure that the language of the participant is adequately captured.

Why did the authors decide that a 3D participatory model is a good way to explore the interviewee's perception of the subsurface. A white 3D box to me is a black box that does not provide much insight. Wouldn't an ordinary whiteboard have been sufficient? What about some transparent 3D-computer models? Minor comments: Line 37: Change "need to educated" to "need to be educated". Figure 2: Increase font size. Figure 5: The grey filled area is hard to see. Why not use a classic pie chart?

---

## Author Comment (AC1) · 11 Apr 2016

Thank you very much for your comments, which I will address below:

1. Thank you for this comment – I have attempted to address the issue of language, particularly the use of heuristics and snowball sampling. 2. Yes I agree, and an example of how poor communication can lead to misconceptions has been added: p2 line 70-76 3. Thank you for this comment, some additional detail has been added to clarify the meaning of this diagram: p5 lines 192-196 4. This is a very interesting suggestion, thank you. Although it would be useful to suggest an alternative method for communicators and geologists who don't have the time to complete the full method, the central aim of this paper is to make a case for using the mental models method to improve the quality and relevance of communication between both expert and non-expert parties. As such I don't feel that I can offer a shorter or more compressed version of this method without doing a disservice to the process.

Additional comments:

P1 line 32 – yes, this has been altered P2 line 44 – yes, a sentence has been added to clarify the meaning of heuristics (line 46-47) P2 line 77 – yes, an example of the messages have been added (line 97-101) P7 line 212 – yes, additional information has been added to clarify (line 248) P7 line 224 – no, a single questionnaire is produced, additional information has been added to clarify (line 257-262) P8 line 252 – hydrological interactions were chosen because they were unexpected, additional information has been added to clarify (line 309-311) P8 line 260 – yes, this has been altered P9 line 287 – yes an additional sentence has been added to provide more information about snowball sampling (line 353-357) P13 line 403 – yes, this is an error – there is an image for 3e, but this reference does not refer to it (also this figure has been changed to Figure 4) P 16 line 529 – yes, thank you, this was confusing – it has been altered (line 622)

Figure 1 – yes, this has been altered (Figure 1)

Figure 3 – yes, this has been altered (is now Figure 4)

Please also note the supplement to this comment:
http://www.hydrol-earth-syst-sci-discuss.net/hess-2015-542/hess-2015-542-AC1-supplement.pdf
* * *
Involuntary
Unknown
Uncontrollable nuclear
power

2.00

food colouring
food preservatives pesticides

1.50

spray cans

1.00

antibiotics
contraceptives          X-rays

0.50

commercial
aviation vaccination          surgery

Not certain to be fatal          -1.00     -0.50          0.50     1.00          Certain to be fatal
Common          Dread
Chronic          Catastrophic railroads
electric power
motor vehicles          general aviation
home appliances          construction          handguns
football          -0.50
smoking
power          bicycles          alcoholic          motorcycles
mowers          beverages          police work
firefighting
hunting
skiing          -1.00
swimming
mountain climbing

Voluntary
Known
Controllable

**Fig. 1.**

a)

b)

c)

d)

**Fig. 2.**

**Fig. 3.**
Interactive
comment

[Figure]

**Fig. 4.**
[Figure]

Fig. 5.

POLLUTION/WASTE GROUND, ENGINE HOUSES, HERITAGE/HISTORY, UNIQUE, LANDSCAPE, AGRICULTURE, INDUSTRY, EMPLOYMENT, CULTURE/COMMUNITY, ECONOMY, FLOODING, WATER, SURFACE, RELATIONSHIP, SOIL-ROCK BOUNDARY, MINING, FEAR, SUBSURFACE, SHAFTS/TUNNELS/UNDERGROUND RIVERS, MINERALS, ANTHROPOCENTRIC, LAYERS, ANGLED DEPOSITS, ROCK TYPES, PROPERTIES OF ROCK, FORMATION OF MINERALS, EARTH'S STRUCTURE, STATE OF ROCK, PERIPHERAL SENSES

**3D THINKING**

**SCALE**

**TECHNICAL AND LOCAL TERMS**

[revised manuscript text omitted]

.

Figure 34.

[Figure]

Figure 45.

[Figure]

Figure 56.

[Figure]

Figure 67.

---

## Author Comment (AC2) · 11 Apr 2016

Thank you very much for your comments which I will try and address below:

In response to the statement about the influence of formal education as providing a 'base level of knowledge that scientists can relate to': I agree, formal education does provide data that the non-expert participant will use in decision making that may be considered more familiar to a scientist, but it is unlikely that that information from a formal educational background will not also be influenced by the participant's own perceptions of their environment. Thus formal educational experience (to whatever level or degree of participation) is included within the broad description of the inherent cultural and social influences that control participant interpretation of new scientific data. Additional detail has been added to clarify this point (line 93).

[Figure]

In response to the statement about the use of analogy: As the mental models method allows for participants to share intuitive theories in their own words during the interview stage, analogies are included in the model, however the method of the interviews also allows for the interviewer to probe certain statements in more detail if required and as such it is possible to discover if the analogy is covering another perception, or if the analogy represents an actual concept for the participants. As such, the use of analogy is not considered directly relevant to this study, as they are either exposed or incorporated into the model itself. However, addition detail has been added to clarify this point (line 251-252).

In response to the comment for clarification of the expert dismissal of the relevance of lay-knowledge: the sentence has been clarified to preclude individual communications, which do often value local knowledge, and clarify the classification of the non-expert approach as inappropriate in relationship to the study done by Johnson (2008) examining public participation in watershed modelling (line 154-159).

In response to the question about the place of deductive reasoning in decision making: I agree that there are other types of reasoning that are involved with decision making and have clarified the sentence to make this more clear (line 176-178).

In response to the query for more data on how the semi-structured interviews and quantitative questionnaires have been designed: yes, additional information has been provided (line 359-374).

In response to the question about the decision to employ a 3D participatory model: additional detail has been provided to clarify the inclusion of the model (line 271-276). Additionally the use of a computer model would not be suitable for issues of practicality.

Minor comments:

Line 37 change : yes, this has been altered (line 39) Figure 2: yes, the font size has been increased as much as possible (Figure 2)

Please also note the supplement to this comment:
http://www.hydrol-earth-syst-sci-discuss.net/hess-2015-542/hess-2015-542-AC2-supplement.pdf

———————————————————
[Figure]

**Fig. 1.**

**Fig. 2.**

[Figure]

**Fig. 3.**

[Figure]

**Fig. 4.**

[Figure]

Fig. 5.

POLLUTION/
WASTE GROUND

ENGINE HOUSES

HERITAGE/
HISTORY

UNIQUE

LANDSCAPE

AGRICULTURE

INDUSTRY

EMPLOYMENT

CULTURE/
COMMUNITY

ECONOMY

SURFACE

RELATIONSHIP

SUBSURFACE

FLOODING

WATER

SOIL-ROCK
BOUNDARY

MINING

FEAR

SHAFTS/TUNNELS/
UNDERGROUND RIVERS

MINERALS

ANTHROPOCENTRIC

LAYERS

ANGLED
DEPOSITS

ROCK TYPES

FORMATION OF
MINERALS

PROPERTIES
OF ROCK

STATE OF ROCK

EARTH'S
STRUCTURE

PERIPHERAL
SENSES

3D THINKING

SCALE

TECHNICAL AND LOCAL TERMS

[Figure]

Fig. 6.

**'Water cannot flow through solid rock'**

28.6% agree    49.5% disagree    21.8% don't know

**Fig. 7.**

**Supplement:**

[revised manuscript text omitted]

.

Figure 34.

[Figure]

Figure 45.

[Figure]

Figure 56.

[Figure]

Figure 67.